

# Optimization of Sulfate Aerosol Hygroscopicity Parameter in WRF-Chem version (3.8.1)

Ah-Hyun Kim[1], Seong Soo Yum[1], Dong Yeong Chang[1], and Minsu Park[1]

[1]Department of Atmospheric Sciences, Yonsei University, Seoul, postal 03722, Korea

*Correspondence to*: Seoung Soo Yum (ssyum@yonsei.ac.kr)

**Abstract.** A new sulfate aerosol hygroscopicity parameter ($\kappa_{SO_4}$) parameterization is suggested that is capable of considering the two major sulfate aerosols, $H_2SO_4$ and $(NH_4)_2SO_4$, using the molar ratio of ammonium to sulfate (R). An alternative $\kappa_{SO_4}$ parameterization method is also suggested that utilizes typical geographical distribution patterns of sulfate and ammonium, which can be used when ammonium data is not available for model calculation. Using the Weather Research and Forecasting

coupled with Chemistry model (WRF-Chem), impacts of different $\kappa_{SO_4}$ parameterizations on cloud microphysical properties and cloud radiative effects in East Asia are examined. Comparisons with the observational data obtained from an aircraft field campaign suggest that the new $\kappa_{SO_4}$ parameterizations simulate more reliable aerosol and CCN concentrations, especially over the sea in East Asia than the original $\kappa_{SO_4}$ parameterization in WRF-Chem that assumes sulfate aerosols as $(NH_4)_2SO_4$ only. With the new $\kappa_{SO_4}$ parameterizations, the simulated cloud microphysical properties and precipitation became significantly

different, resulting in greater cloud albedo effect of about -1.5 W m$^{-2}$ in East Asia than that with the original $\kappa_{SO_4}$ parameterization. The new $\kappa_{SO_4}$ parameterizations are simple and readily applicable to numerical studies of investigating the impact of sulfate aerosols in aerosol-cloud interactions without additional computational expenses.

## 1 Introduction

Aerosols impact global climate by directly scattering and absorbing radiation. Aerosols also play an important role

as potential cloud condensation nuclei (CCN). Increases in CCN number concentration could increase cloud optical depth, suppress local precipitation, and prolong cloud lifetime (Twomey, 1974; Albrecht, 1989). Therefore, the aerosol induced changes of cloud microphysical property can alter the Earth's radiation budget and hydrological cycle. Such aerosol-cloud interactions possibly cause the greatest uncertainty in the estimation of climate forcing due to their complexity (Myhre et al., 2013). Understanding the role of aerosols as CCN (CCN activation) is therefore important for predicting future climate. CCN

activation depends on the chemical and physical properties of aerosols (Köhler 1963; Abdul-Razzak et al., 1998; Dusek et al., 2006; Fountoukis and Nenes, 2005; Khvorostyanov and Curry, 2009; Ghan et al., 2011). Soluble aerosol species have high potential to become CCN and differences in aerosol solubility could exert a considerable impact on CCN activation (Nenes et al., 2002; Kristjánsson 2002).

Sulfate aerosols are one of the major components of natural and anthropogenic aerosols, contributing to a large

portion of the net radiative forcing due to aerosol-cloud interactions (Boucher et al, 2013). They are highly soluble and thereby



easily activated to become cloud droplets. Recently, Zelinka et al. (2014) estimated that the contribution of sulfate aerosols to the net effective radiative forcing from aerosol-cloud interaction (ERFaci) is about 64%. Sulfate aerosols are mainly present as sulfuric acid ($H_2SO_4$) and ammonium sulfate (($NH_4$)$_2SO_4$) in the atmosphere (Charlson and Wigley, 1994), but have very different hygroscopicity parameter ($\kappa$) that represents the water affinity of aerosols and determines the efficiency of CCN

activation (Petters and Kreidenweis, 2007). Despite the importance of sulfate aerosols in the estimation of ERFaci, many atmospheric models treat sulfate aerosols simply assuming that they have a single $\kappa_{SO_4}$ value (Ackermann et al., 1998; Stier et al. 2006; Pringle et al., 2010; Mann et al., 2011; Chang et al., 2017; Tegen et al., 2019).

Especially in East Asia, distribution of $\kappa_{SO_4}$ value could vary significantly because the sulfur dioxide and ammonia are emitted from inland China on a massive scale (Kurokawa et al., 2013; Qu et al., 2016; Kang et al., 2016; Liu et al., 2017)

and the distribution of $H_2SO_4$ and ($NH_4$)$_2SO_4$ are closely related to the emissions and chemical reactions of sulfur dioxide and ammonia. Sulfur dioxide is oxidized to $H_2SO_4$ and then neutralized to form ($NH_4$)$_2SO_4$ by ammonia. Generally, sulfur dioxide is released from industries and from the sea surface, and ammonia is discharged from livestock and farmland. For this reason, the ratio of ammonium to sulfate is observed to decrease as the distance from the land increases (Fujita et al., 2000; Paulot et al., 2015; Kang et al., 2016; Liu et al., 2017). Thus, applying a single hygroscopicity parameter for all sulfate aerosols in

atmospheric models can lead to uncertainty in quantifying CCN activation, particularly in East Asia.

This study proposes a new $\kappa_{SO_4}$ parameterization that aims at simultaneously considering the two major sulfate aerosols, i.e., ($NH_4$)$_2SO_4$ and $H_2SO_4$ in WRF-Chem (the Weather Research and Forecasting model coupled with chemistry model). First, we describe the calculation of $\kappa$ for different size modes of aerosols and suggest a new parameterization of $\kappa_{SO_4}$. The performance of the new $\kappa_{SO_4}$ parameterization in estimating the effects of aerosol-cloud interactions is examined for the

domain of East Asia. The model results are compared with the aircraft measurement data obtained during the Korea–US Air Quality campaign (KORUS-AQ, Al-Saadi et al., 2016). Finally, we address the effects of the new $\kappa_{SO_4}$ parameterizations in simulating (or calculating) cloud microphysical properties and cloud radiative effects in East Asia.

## 2 Model description

### 2.1 The WRF-Chem model

The WRF-Chem version 3.8.1 is designed to predict mesoscale weather and atmospheric chemistry (Grell et al., 2005; Fast et al., 2006; Skamarock et al., 2008; Peckham et al., 2011). The aerosol size and mass distributions are calculated with the Modal Aerosol Dynamics Model for Europe (MADE; Ackermann et al., 1998) that includes three log-normal distributions for Aitken, accumulation, and coarse mode particles. MADE considers the new particle formation process of homogeneous nucleation in $H_2SO_4$ and $H_2O$ system (Wexler et al., 1994; Kulmala et al., 1998). MADE treats inorganic chemistry systems

as the default option and organic chemistry systems as coupling options. Inorganic chemistry systems include the chemical reactions of three inorganic ionic species, i.e. $SO_4^{-2}$, $NO_3^-$ and $NH_3^+$ (Ackermann et al., 1998). The secondary organic aerosol



model (SORGAM), an optional model to calculate secondary organic aerosol chemistry processes (Schell et al., 2001), is coupled to MADE (MADE/SORGAM). MADE/SORGAM treats atmospheric aerosols as an internal mixture of sulfate, nitrate, ammonium, organic carbon (OC), elemental carbon (EC), sea salt, and dust aerosols. Additionally, gas phase chemical
processes are calculated in Regional Acid Deposition Mechanism version 2 (RADM2, Chang et al., 1989). RADM2 simulates the concentrations of air pollutants, including inorganic (14 stable, 4 reactive and 3 abundant stable) and organic (26 stable and 16 peroxy radicals) chemical species.

For the microphysics calculation, we use the CCN activation parameterizations (Abdul-Razzak and Ghan, 2000, hereafter ARG) and Morrison double-moment microphysics scheme (Morrison et al., 2009). The CCN activation is determined
by meteorological factors (e.g., updraft) and physicochemical properties of aerosols based on the assumption of internally well-mixed aerosols. A detailed model designs for the modelling studies of aerosol–cloud interactions in WRF-Chem can be found in Gustafson et al. (2007), Chapman et al. (2009), Grell et al. (2011), and Baró et al. (2015).

For the physics parameterization, we use the following configurations: Rapid and accurate Radiative Transfer Model for GCMs (RRTMG) for the shortwave and longwave radiative transport processes (Iacono et al., 2008); Yonsei University
scheme (YSU scheme) for the atmospheric boundary layer processes (Hong et al., 2006) ; the Unified NOAH (NCEP Oregon State University, Air Force, and Hydrologic Research Lab's) Land Surface Model for land surface processes (Tewari et al., 2004).

## 2.2 Calculation of the hygroscopicity parameter

The CCN activation parameterization is based on the Köhler theory that is described with the water activity and the
surface tension of the solution droplets. The water activity is estimated from detailed information of aerosols such as Van't Hoff factor, osmotic coefficient, molecular weight, mass, and density of aerosols. If aerosol chemical information is fully provided, CCN activation could be almost accurately calculated using the Köhler theory (Raymond and Pandis, 2003). However, it requires high computational expenses (Lewis, 2008). Petters and Kreidenweis (2007) proposed a single quantitative measure of aerosol hygroscopicity, known as hygroscopicity parameter ($\kappa$). This method does not require detailed
information of aerosol chemistry and thereby reduces computational cost when calculating the water activity. For this reason, $\kappa$ values are applied in many observational, experimental, and numerical studies (Zhao et al., 2015; Shiraiwa et al., 2017; Chang et al., 2017; Gasteiger et al., 2018). $\kappa$ can be determined separately for the three log-normal modes (Aitken, accumulation, and coarse modes). That is, $\kappa_i$ is the volume-weighted average of $\kappa_j$ for mode $i$:

$$\kappa_i \equiv \sum_{j=1}^{J} \varepsilon_{ij} \kappa_j, \tag{1}$$

where $\varepsilon_{ij}$ is the volume ratio of chemical j in mode $i$ (=$V_{ij}/V_{tot,i}$, $V_{tot,i} = \sum_{j=1}^{J} V_{ij}$, and $V_{ij}$ is the volume of chemical $j$ in mode
$i$) and $\kappa_j$ is the individual hygroscopicity parameter for chemical $j$. Eq. (1) is calculated with the assumption of the temperature, 298.15 K. The upper bound of $\kappa$ is around 1.40 (Petter and Kreidenweis, 2007).



### 2.3 Limitation of previous $\kappa_{SO_4}$ parameterizations

CCN activation is affected by $\kappa$ values (e.g., Nenes et al., 2002; Kristjánsson 2002). H$_2$SO$_4$ has more than two times

higher $\kappa$ value than (NH$_4$)$_2$SO$_4$, i.e., 1.19 for $\kappa_{H_2SO_4}$ and 0.53 for $\kappa_{(NH_4)_2SO_4}$ (Clegg and Wexler, 1998; Petters and Kredenweis 2007; Good et al., 2010). Such large disparities of $\kappa_{SO_4}$ between different sulfate species could cause a large variability in the estimation of ERFaci. However, many aerosol modules simplify the physical and chemical characteristics of aerosols, and often neglect some chemical species (Kukkonen et al., 2012; Im et al., 2015; Bessagnet et al., 2016). Sulfate aerosols are usually prescribed as a single species of either H$_2$SO$_4$ or (NH$_4$)$_2$SO$_4$. Some models consider H$_2$SO$_4$ as the representative sulfate

aerosol when the neutralization reaction between H$_2$SO$_4$ and ammonia is not considered or when only the binary sulfuric acid–water nucleation is considered (e.g., Mann et al., 2011; Kulmala et al., 1998; Korhonen et al., 2008; Stier et al., 2006; Wexler et al., 1994; Kazil and Lovejoy 2007). Some other models consider (NH$_4$)$_2$SO$_4$ as the representative sulfate aerosol when studying aerosol-CCN closure (e.g., VanReken et al., 2003) or when including the ternary sulfuric acid–ammonia–water nucleation process or the neutralization reaction between sulfate and ammonia (Kulmala et al., 2002; Napari et al., 2002;

Elleman and Covert, 2009; Watanabe et al., 2010; Grell et al., 2005). To reduce the uncertainty of ERFaci, more speciated $\kappa_{SO_4}$ parameters need to be utilized in the calculation of cloud droplet activation process, at least, for the two main sulfate aerosols i.e., H$_2$SO$_4$ and (NH$_4$)$_2$SO$_4$. Here, we suggest a new method of representing $\kappa_{SO_4}$ that considers both H$_2$SO$_4$ and (NH$_4$)$_2$SO$_4$ using the molar ratio of NH$_4^+$ to SO$_4^{2-}$. We also suggest an alternative method that utilizes the spatial distribution of $\kappa_{SO_4}$, based on the distinct distribution patterns of NH$_4^+$ and SO$_4^{2-}$ over land and sea.

### 110  2.4 New parameterization of $\kappa_{SO_4}$

H$_2$SO$_4$ is completely neutralized as (NH$_4$)$_2$SO$_4$ when ammonia is abundant (Seinfeld and Pandis, 2006). During the neutralization process of H$_2$SO$_4$, one mole of SO$_4^{2-}$ takes up two moles of NH$_4^+$ and forms one mole of (NH$_4$)$_2$SO$_4$. Here, the assumption is that ammonia neutralizes SO$_4^{2-}$ ions prior to nitrate ions (Seinfeld and Pandis, 2006) and sulfate aerosols appear only in the form of H$_2$SO$_4$ and (NH$_4$)$_2$SO$_4$. In the calculation of $\kappa_{SO_4}$, the proportion of H$_2$SO$_4$ and (NH$_4$)$_2$SO$_4$ is determined

using the ammonium to sulfate molar ratio $R$ ($=n_{NH_4^+}/n_{SO_4^{2-}}$, $n_{NH_4^+}$ is the molar concentration of NH$_4^+$ ions and $n_{SO_4^{2-}}$ is the molar concentration of SO$_4^{2-}$ ions). Generally, sulfate aerosols are completely neutralized as (NH$_4$)$_2$SO$_4$ at high $R$ conditions ($R>2$), and partially neutralized at low $R$ conditions ($R<2$) (Waggoner et al., 1983; Fisher et al., 2011). Using $R$ and the Zdanovskii–Stokes–Robinson relationship (i.e., $V_d = V_w + V_{tot}$, $V_{tot} = \sum_{j=1}^{J} V_j$, where $V_d$ is the droplet volume, $V_w$ is the volume of water, and $V_j$ is the volume of the chemical $j$), a representative $\kappa_{SO_4}$ is defined as:

$$\kappa_{SO_4} = \varepsilon_{H_2SO_4}\kappa_{H_2SO_4} + \varepsilon_{(NH_4)_2SO_4}\kappa_{(NH_4)_2SO_4}, \tag{2}$$

where $\varepsilon_{H_2SO_4}$ is the volume fraction of H$_2$SO$_4$ in the total volume of sulfate aerosols ($=V_{H_2SO_4}/V_{SO_4}$; $V_{H_2SO_4}$ is the volume concentration of H$_2$SO$_4$, $V_{SO_4}$ is the total volume concentration of sulfate aerosols), and $\varepsilon_{(NH_4)_2SO_4}$ is calculated likewise for





$(NH_4)_2SO_4$ [$=V_{(NH_4)_2SO_4}/V_{SO_4}$; $V_{(NH_4)_2SO_4}$ is the volume concentration of $(NH_4)_2SO_4$]. In this study, we use 1.19 and 0.53 to represent $\kappa_{H_2SO_4}$ and $\kappa_{(NH_4)_2SO_4}$, respectively (Clegg and Wexler, 1998; Petters and Kredenweis 2007; Good et al., 2010). The volume fractions of $H_2SO_4$ and $(NH_4)_2SO_4$ are calculated as:

i)    if $R \leq 0$, then, $\varepsilon_{H_2SO_4} = 1$ and $\varepsilon_{(NH_4)_2SO_4} = 0$,

ii)    if $0 < R < 2$, then,

$$\varepsilon_{H_2SO_4} = \frac{\left[\left(1-\frac{R}{2}\right) \times n_{SO_4^{2-}}\right] \times \frac{m_{H_2SO_4}}{\rho_{H_2SO_4}}}{V_{SO4}} \text{ and } \varepsilon_{(NH_4)_2SO_4} = \frac{\left(\frac{R}{2} \times n_{SO_4^{2-}}\right) \times \frac{m_{(NH_4)_2SO_4}}{\rho_{(NH_4)_2SO_4}}}{V_{SO4}},$$ (3)

iii)    if $R > 2$, then $\varepsilon_{H_2SO_4} = 0$ and $\varepsilon_{(NH_4)_2SO_4} = 1$,

where $m$ and $\rho$ indicate the molar mass and density of the specific chemical species, respectively. As a result, sulfate aerosols are treated as $(NH_4)_2SO_4$ when $R$ is greater than two ($R>2$) and as $H_2SO_4$ when $R$ is zero ($R=0$). This method is applicable to the models that considers both $NH_4^+$ and $SO_4^{2-}$ ions. If $NH_4^+$ data is not available in a model, we suggest an alternative method to represent $\kappa_{SO_4}$, based on the typical geographical distribution pattern of sulfate aerosols available from observations as discussed below.

130         Observational studies show the distinctly different distribution patterns of the two dominant sulfate aerosol species, i.e., $(NH_4)_2SO_4$ over land and $H_2SO_4$ over sea (Fujita et al., 2000; Paulot et al., 2015; Kang et al., 2016; Liu et al., 2017). Such distribution pattern is related to the sources of sulfate and ammonium. In general, sulfate aerosols are emitted from land and sea, while ammonium is mostly produced from lands. Sulfur dioxide is produced from fossil fuel combustions, volcanic eruptions and dimethyl-sulfide (DMS) via air-sea exchanges, and then forms sulfate aerosols (Aneja 1990; Jardin et al., 2015).

Wind transportations of pollutants could also cause high concentration of sulfate aerosols over the sea (Liu et al., 2008). On the other hand, ammonium is emitted from livestock, fertilizer, and vehicles (Sutton et al., 2013; Paulot et al., 2014; Bishop et al., 2015; Liu et al., 2015; Stritzke et al., 2015), and therefore is concentrated mostly on lands. Ammonium is usually not abundant enough to fully neutralize $H_2SO_4$ in the marine boundary layer (Paulot et al., 2015; Ceburnis et al., 2016). Therefore, when the ammonium information is not available, the $\kappa_{SO_4}$ can be alternatively estimated with the consideration of the land

and sea fractions as:

$$\kappa_{SO_4} = f \times \kappa_{SO_4,land} + (1-f) \times \kappa_{SO_4,sea},$$ (4)

where $f$ represents the fraction of land at each grid point; unity means entire land, and zero means entire sea, and the value in-between represents the fraction of land at the grid points in the coastal areas. $\kappa_{SO_4,land}$ and $\kappa_{SO_4,sea}$ represent $\kappa_{SO_4}$ over land and sea, respectively (i.e., $\kappa_{SO_4,land} = \kappa_{(NH_4)_2SO_4} = 0.53$ and $\kappa_{SO_4,sea} = \kappa_{H_2SO_4} = 1.19$).



## 3 Experimental setup

Model simulations are carried out for thirty-six days from 0000 UTC 10 May to 0000 UTC 15 June 2016 and the first five days are used as a spin-up time. Observational data for sulfate aerosols and CCN during this period were obtained from the KORUS-AQ campaign and they indicated that sulfate aerosols were widely distributed throughout East Asia due to the stagnation of high-pressure systems and the transportation of pollutants from China. The domain covers East Asia (i.e., 2,700 km × 2,700 km; 20°N–50°N, 105°E–135°E) with 18 km grid spacing and 50 vertical levels from the sea level pressure to 100 hPa. The initial and boundary conditions are provided by the National Center for Environment Prediction–Climate Forecast System Reanalysis (NCEP–CFSR) (Saha et al., 2014). The 4DDA (4 Dimensional Data Assimilation) analysis nudging is used. Anthropogenic emission inventories are obtained from Emissions Database for Global Atmospheric Research–Hemispheric Transport of Air Pollution (EDGAR–HTAP; Janssens-Maenhout et al., 2015). Natural source emission inventories adopt the Model of Emissions of Gases and Aerosols from Nature (MEGAN; Guenther et al., 2006).

We conduct four simulations with different $\kappa_{SO_4}$ parameterizations: 1) AS uses a single $\kappa_{SO_4}$ of 0.53 (i.e., $\kappa_{(NH_4)_2SO_4}$), assuming that all sulfate aerosols are completely neutralized by ammonium, which is a default setting in WRF-Chem; 2) SA uses a single $\kappa_{SO_4}$ of 1.19 (i.e., $\kappa_{H_2SO_4}$) assuming that all sulfate aerosols are H₂SO₄; 3) RA applies the new $\kappa_{SO_4}$ parameterization that calculates the volume weighted mean $\kappa_{SO_4}$ by using the molar ratio of ammonium to sulfate ($R$) (i.e., eq. 2); and 4) LO adopts different $\kappa_{SO_4}$ for land and sea, assuming that sulfate aerosols are completely neutralized as (NH₄)₂SO₄ over land and are H₂SO₄ only over sea (i.e., eq. 4).

## 4 Results and discussions

### 4.1 Distribution of sulfate and ammonium

The simulated sulfate and ammonium distributions are compared with the observational data measured onboard the NASA DC-8 aircraft during the KORUS–AQ campaign (https://www-air.larc.nasa.gov/missions/korus-aq/), in and around the Korean Peninsula in May and June of 2016. The mass concentration of sulfate and ammonium were obtained by the method described in Dibb et al. (2003).

In Fig. 1, the mass concentration of sulfate and ammonium simulated by AS are compared with the KORUS-AQ aircraft observations (OBS) following the flight track. The simulated sulfate shows a positive bias but has a high temporal correlation with OBS (r=0.78). The simulated ammonium is less biased than sulfate but indicates a moderate temporal correlation with OBS (r=0.58). Overall, it seems reasonable to state that the WRF-Chem model can calculate the distribution of sulfate aerosols well enough.

Fig. 2 shows 30 days averaged mass concentration of sulfate and ammonium and the molar ratio ($R$) of ammonium to sulfate over the model domain. During the KORUS-AQ campaign period, high-pressure systems often covered East China and the Yellow Sea and that led to stagnating sulfate and ammonium concentrations. However, sulfate and ammonium are





distributed differently due to different sources. Pollutants emitted from the Asian continent are often transported by westerly and southerly winds. Sulfate is highly concentrated in China and the northern part of the Yellow Sea, and DMS emission from the sea also contributes to the formation of sulfate aerosols over the sea. Ammonium is widely distributed throughout China due to the use of fertilizers over farmlands (Van Damme et al., 2014; Paulot et al., 2014; Warner et al., 2017). The concentration
of ammonium is generally low over the sea, but over the northern part of the Yellow Sea it is high due to wind transport.

The distribution of $R$ is associated with the distribution of sulfate and ammonium (Fig. 2). In general, $R$ is high ($R>2$) over the land on account of the high anthropogenic emissions of continental ammonium, and $R$ is low ($R<2$) over the remote seas because the ammonium concentration is small. However, high $R$ is also shown over the Yellow Sea in Fig. 2. This is because the ammonium concentration increases when the westerlies carry continental pollutants over the Yellow Sea during
the simulation period. Based on the distribution of $R$, sulfate aerosols are expected to be almost completely neutralized over land [e.g., $(NH_4)_2SO_4$] and partially neutralized over sea [$(NH_4)_2SO_4+H_2SO_4$].

### 4.2 Distribution of $\kappa$

Fig. 3 shows the average $\kappa$ of the accumulation mode aerosols in AS and the difference between RA and AS, and LO and AS. The accumulation mode is selected because sulfate aerosols are dominant in this mode. AS simulates $\kappa$ values that are
roughly consistent with the observed mean $\kappa$ values in the literature (i.e. $\kappa$ over land is about 0.3 and $\kappa$ over sea is about 0.7; Andreas and Rosenfeld, 2008), but it varies significantly between land and sea. The $\kappa$ over land is expected to be lower than the $\kappa$ over sea because continental aerosols usually include more hydrophobic aerosol species such as black carbon and organic carbon from industries, while maritime aerosols consist mainly of hydrophilic substances, i.e. sea salt and non-sea salt sulfates originated from DMS. The variation of $\kappa$ is also influenced by chemical reactions and meteorological factors, i.e. wind
transportations of aerosols and scavenging of aerosols due to precipitations, as well as gravitational settling.

Compared to AS, RA and LO show pronounced difference of $\kappa$ over sea (Figs. 3b and 3c). That is, RA and LO produce significantly higher $\kappa$ over the sea than AS does because ammonium concentration is not sufficient enough to neutralize sulfate completely over the sea (i.e. R<2). RA predicts slightly higher continental $\kappa$ following the coastal regions than AS because $R$ becomes occasionally low due to the intrusion of maritime air mass that has very low concentration of
ammonium. Maritime $\kappa$ of RA is lower than that of LO because the transportation of continental pollutants increases the portion of ammonium over the Yellow sea.

### 4.3 CCN activation

According to the Köhler theory, changes in $\kappa$ directly influence CCN activation. In this study, the CCN activation rate ($f_{CCN}$) is defined as the ratio of the CCN number concentration at 0.6% supersaturation to the total aerosol number
concentration. Simulated $f_{CCN}$ is compared with the aircraft measurements during the KORUS-AQ campaign (OBS). During this campaign, aerosol and CCN number concentrations were measured by a condensation particle counter (CPC; TSI 3010)





and a CCN counter (CCNC; DMT CCN-100), respectively (Park et al., 2020). The CPC measures the number concentration of aerosols larger than 10 nm in diameter, and the CCNC measures the CCN number concentration at 0.6% supersaturation.

The model simulations well capture the temporal variation of $f_{CCN}$ (r~0.7 for the linear correlation with OBS; Fig. 4).

However, $f_{CCN}$ are underestimated due to the overestimated total aerosol concentration. According to Georgiou et al.(2018), the WRF-Chem coupled with MADE/SORGAM tends to overestimate aerosol number concentrations. Over land, AS, RA and LO simulate similar values of $f_{CCN}$ because continental sulfate aerosols are generally expected to be fully neutralized form of sulfate [i.e. $(NH_4)_2SO_4$]. This was not the case over sea. During KORUS-AQ, the aircraft passed over the Yellow Sea on 22 and 25 May 2016 (blue shades in Fig. 4). For this occasion, LO simulates the highest $f_{CCN}$ over the sea among all simulations

because LO uses the prescribed $\kappa_{SO_4}$ value of $\kappa_{H_2SO_4}$ over sea. RA simulates slightly lower $f_{CCN}$ over the sea because transportation of continental pollutants over the sea can be taken into account, as observed during the KORUS-AQ campaign. The transported air pollutants increase the ammonium concentration over the sea, neutralize $H_2SO_4$, reduce the hygroscopicity of sulfate aerosols, and consequently decrease $f_{CCN}$. Compared to the RA and LO simulations, AS predicts the lowest $f_{CCN}$ because the lowest $\kappa_{SO_4}$ (=$\kappa_{(NH_4)_2SO_4}$) is prescribed over sea as well as over land.

We conducted a reliability test that has been often used to evaluate the performance of air quality models (Kumar et al., 1993). When compared to the observation, RA and LO showed good performances in simulating $f_{CCN}$. That is, RA and LO satisfy the following three criteria suggested by Kumar et al.(1993): 1) the normalized mean squared error below 0.5, 2) the fractional bias (= $2 \times \frac{\overline{obs - sim}}{\overline{obs + sim}}$, *obs* indicates the observed values, *sim* indicates the simulated values, and the bar above the symbols indicate the average) between -0.5 and 0.5, and 3) the ratio of the model values to the observed values (= $sim/obs$)

between 0.5 and 2.0. However, AS does not meet these criteria because it predicts a rather high normalized mean squared error. All in all, RA and LO outperform AS based on the reliability test (Kumar et al., 1993). Between RA and LO, LO seems somewhat closer to observation than RA but the difference is small for these calculations.

The variation of $\kappa_{SO_4}$ almost directly influences the change in column-integrated $f_{CCN}$ (Fig. 5). RA predicts higher $f_{CCN}$ than AS over the coastal land regions because the occasionally very low ammonium concentration lowers $R$ and affects

the CCN activation. Meanwhile, SA prescribes $\kappa_{SO_4}$ value two times as high as AS does and produces about 20% higher $f_{CCN}$.

## 4.4 Cloud microphysical properties

Different $\kappa_{SO_4}$ parameterizations affect simulated cloud microphysical properties. Figure 6 shows the relative differences of the simulated column-integrated cloud droplet number concentration (CDNC) in RA, LO, and SA from AS. All three produce higher $\kappa_{SO_4}$ values than AS, and therefore simulate higher CDNCs. However, the differences in CDNC does not

exactly correspond to the differences in $f_{CCN}$ (Fig. 5) because cloud droplet activation is also affected by in-cloud supersaturation and other meteorological factors. SA simulates higher CDNC than AS over both land and sea, but RA and LO simulate higher CDNC mostly only over sea. RA and LO produce similar CDNC distributions over the Yellow Sea (compare Fig. 6a and Fig. 6b) although RA produces smaller $f_{CCN}$ than LO (compare Fig. 5a and Fig. 5b). As in Moore et al. (2011), the





reason for that may be because the sensitivity on $f_{CCN}$ reduces so much because supersaturation is so high that most aerosols
can act as CCN regardless of their critical supersaturation. That is, the supersaturation over the Yellow Sea is high enough to
activate most aerosols to cloud droplets. Over land, RA simulates higher CDNC (up to 12%) than AS in southeast China and
the Korean Peninsula, but LO simulates CDNC similar to AS. The results of RA seem to be related to the dilution of ammonium
concentrations along the coastal land regions due to the intrusion of maritime air. However, such variation of ammonium
cannot be taken into account in LO.

245       Overall, high CDNCs in RA, LO, and SA (Table 1) result in less precipitation but larger LWP, compared to AS
(Table 2). Precipitation reduction is more pronounced over sea because of larger relative differences in CDNC. These results
agree well with some previous studies; i.e., high CDNCs suppress local precipitations, prolong cloud lifetime, and
consequently increase net LWP, which is known as cloud lifetime effect (Albrecht, 1989). Obviously, SA that assumes sulfate
aerosols are all $H_2SO_4$ particles produces the highest CDNC and also the largest differences in all other properties in Table 2.
Less rainwater in SA than in any other simulations may also imply that precipitation scavenging of aerosols was less efficient
and therefore retaining more aerosols (CCN) to produce more cloud drops and longer cloud lifetime. On average, SA has 103
cm$^{-3}$ more aerosols over sea and 116 cm$^{-3}$ more aerosols over land than LO. These surplus aerosols certainly have potential to
simulate higher number of CCN in SA than LO.

      For the same LWP condition, high CDNC induces small effective radii ($r_e$). RA, LO, and SA simulate smaller $r_e$ than
AS and the maximum difference of $r_e$ amounts to be 1.46 μm, 1.38 μm, and 1.48 μm, respectively. However, the domain
averaged differences of $r_e$ are not as substantial as the differences of other cloud microphysical properties (Table 2). This may
be related to somewhat larger LWPs in RA, LO and SA than AS and the sufficient water supply during droplet growth. All
simulations in this study have high water vapor path (WVP) conditions (WVP> 30 kg m$^{-2}$) throughout the whole domain.
According to Qiu et al. (2017), cloud droplets have small competition for water vapor and high chance of collision-coalescence
under high WVP conditions (i.e., WVP>1.5 cm or 15 kg m$^{-2}$). If LWP is similar, $r_e$ difference could be larger among the
simulations than that are shown herein.

**4.5 Cloud radiative effects**

      Cloud microphysical properties determine cloud optical properties and therefore control the cloud radiative effects.
For a fixed LWP, high CDNC is usually associated with small $r_e$ but large cloud optical thickness. Then optically thick clouds
reflect more sunlight and strengthen cloud radiative cooling effect at the top of the atmosphere (TOA), which is known as
cloud albedo effect (Twomey, 1974). We calculate the cloud radiative effect at the TOA (CRE) by subtracting the clear sky
downward radiation from the net all-sky downward radiation (including clouds) (Hartmann, 2015).

      RA, LO, and SA simulate optically thicker clouds that reflect more sunlight and exert stronger cooling effects at the
TOA than AS (Fig. 7). For the domain average, the differences of CRE for RA, LO, and SA from AS amount to be about -1.7,
-1.5, and -2.1 W m$^{-2}$, respectively. These differences are most pronounced over sea (Figs. 7b, 7c and 7d). Such pronounced
difference over sea may be affected by large cloud fraction around the East China Sea due to the East Asian Summer Monsoon





(Pan et al., 2015). That is, large cloud fraction exerts large CRE cooling, so the impact of the new parameterization of $\kappa_{SO_4}$ on CRE could be substantial under large cloud fraction condition. Note that in the latitude band of 25-28 °N, CRE is similar over the land and over the sea in AS (Fig. 7a) but the CRE differences between RA, LO and SA, and AS are much higher over the

sea than over the land (Fig. 7b, 7c and 7d). Such enhanced cooling effect over sea can be explained by increases in CDNC (Fig. 6) and somewhat by increases in LWP (Table 2). According to some previous studies, the contribution of CDNC and LWP to CRE could be larger than 56% (Sengupta et al., 2003; Goren and Rosenfeld, 2014).

## 5 Summary and conclusions

This study introduced a new hygroscopicity parameterization method of sulfate aerosols in the WRF-Chem model and demonstrates the impacts of different $\kappa_{SO_4}$ parameterization on simulating cloud microphysical properties in East Asia. The new $\kappa_{SO_4}$ parameterization considered the composition effect of $H_2SO_4$ and $(NH_4)_2SO_4$, using the molar ratio of ammonium to sulfate, $R$. We also suggest an alternative $\kappa_{SO_4}$ parameterization: $\kappa_{(NH_4)_2SO_4}$ for land and $\kappa_{H_2SO_4}$ for sea, which utilizes the information of the typical observed geographical distribution of sulfate aerosols, in case when ammonium data are

not available. The performance of the new $\kappa_{SO_4}$ parameterizations was evaluated by comparing with observational data obtained from a field campaign in East Asia and it was demonstrated that the new $\kappa_{SO_4}$ parameterizations could produce more reliable aerosol and CCN concentrations than the previous method that uses a single $\kappa_{SO_4}$ (i.e., $\kappa_{(NH_4)_2SO_4}$).

The effect of the new $\kappa_{SO_4}$ parameterizations is indicated as substantially different cloud microphysical properties especially over the sea (about 20% increases in CDNC). The increases of CDNC suppress local precipitation, prolong cloud

lifetime, and consequently reflect more sunlight, i.e., more cooling effect (about 1.5 W m$^{-2}$) than the simulation with the original $\kappa_{SO_4}$ parameterization in WRF-Chem that assumes $\kappa_{SO_4} = \kappa_{(NH_4)_2SO_4}$ for all sulfate aerosols. These results indicate that the estimated cloud radiative forcing due to aerosol–cloud interactions can vary significantly with different $\kappa_{SO_4}$ parameterizations.

The importance of oceanic sulfate aerosols on radiative forcing is highlighted in recent studies which suggested that

DMS (precursor of oceanic sulfate aerosols) emissions significantly contribute to the total radiative forcing due to aerosol–cloud interactions (Carslaw et al., 2013; Yang et al., 2017). The new $\kappa_{SO_4}$ parameterizations could be more appropriate for studying the effects of oceanic sulfate aerosols on climate, compared to other approaches that use a single $\kappa_{SO_4}$ (i.e., $\kappa_{(NH_4)_2SO_4}$) or use an empirical relationship between $(NH_4)_2SO_4$ and CCN to calculate CCN activation (Boucher and Anderson, 1995; Kiehl et al., 2000). All in all, the new $\kappa_{SO_4}$ parameterization is capable of considering the variation of $\kappa_{SO_4}$ and simulates more

reliable results, compared to the previous method using a single $\kappa_{SO_4}$ value in the calculation of cloud microphysical properties. Many atmospheric models neglect the differences of hygroscopicity between $H_2SO_4$ and $(NH_4)_2SO_4$ for simplicity. However, this could result in large uncertainties in estimating CRE, especially in East Asia, as demonstrated in our results.



Therefore, we propose this new parameterization of $\kappa_{SO_4}$ that considers both of the two dominant sulfate aerosols, $H_2SO_4$ and $(NH_4)_2SO_4$, for investigating the effects of sulfate aerosols on climate, especially for East Asia, which shows
distinctly different emission patterns over land and sea. The new parameterizations are applicable to calculate CCN activation without additional treatments of the chemical reactions and computational expenses. The new parameterization introduced in this study is expected to work effectively in the domain where land and sea are almost evenly distributed, or in the regions with varying distribution of ammonium to sulfate molar ratio. However, we tested the performance of the new $\kappa_{SO_4}$ parameterization only in East Asia due to the limited amount of observation data available to validate the performance of CCN
activation. Therefore, further studies are needed for different regions where observation data are available to confirm the reliability of our new parameterization.

**Code availability.**

The original WRF-Chem v3.8.1 source code is available at https://www2.mmm.ucar.edu/wrf/users/download/get_sources.html (Grell et al., 2005; Fast et al., 2006; Skamarock et al.,
2008; Peckham et al., 2011). The optimized sulfate aerosol hygroscopicity parameter code is available at https://zenodo.org/record/3899838#.Xusv82ozZTY; DOI: 10.5281/zenodo.3899838.

**Data availability.**

The National Center for Environment Prediction–Climate Forecast System Reanalysis (NCEP–CFSR) data for initial and
boundary conditions are obtained from https://rda.ucar.edu/datasets/ds093.1/ (Saha et al., 2014). Anthropogenic emission inventories are obtained from Emissions Database for Global Atmospheric Research–Hemispheric Transport of Air Pollution (EDGAR–HTAP; Janssens-Maenhout et al., 2015). The data measured from DC-8 aircraft during the KORUS–AQ campaign is available at https://www-air.larc.nasa.gov/missions/korus-aq/.

**Author contributions.**

AK constructed the idea, designed the optimization method and wrote the first draft of the manuscript. SSY acquired funding and supervised the whole study, and edited the manuscript. DYC participated in the construction of the idea and development of the optimization method, and edited the manuscript. MP provided the KORUS-AQ Champaign data and edited the manuscript.



**Acknowledgement**

This work was funded by the Korea Meteorological Administration Research and Development Program under Grant KMI2018-03511.

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





**Tables**

**Table 1.** Domain averaged differences ($= \frac{sensitivity\ simulation\ -AS}{AS} \times 100\%$) of the CCN activation fraction at 0.6% supersaturation ($f_{CCN}$) and cloud droplet number concentration (CDNC) in percent. The data are averaged from 0000 UTC 15 May to 0000 UTC 15 June.

|  | RA–AS | | LO–AS | | SA–AS | |
|---|---|---|---|---|---|---|
|  | Land | Ocean | Land | Ocean | Land | Ocean |
| $f_{CCN}$ (%) | 6 | 13 | 1 | 19 | 18 | 22 |
| CDNC (%) | 7 | 20 | 1 | 21 | 14 | 24 |

**Table 2.** Domain averaged water budgets of AS and their differences from other simulations. The data are averaged from 0000 UTC 15 May to 0000 UTC 15 June. Rainwater in this study refers to the liquid phase of water that has a potential of becoming rainfall in the model.

|  | AS | | RA–AS | | LO–AS | | SA–AS | |
|---|---|---|---|---|---|---|---|---|
|  | Land | Ocean | Land | Ocean | Land | Ocean | Land | Ocean |
| Rainwater (g m$^{-2}$) | 21.6 | 39.4 | -0.32 | -0.60 | -0.02 | -0.64 | -0.52 | -0.73 |
| LWP (g m$^{-2}$) | 45.7 | 78.4 | 0.40 | 1.41 | 0.08 | 1.45 | 0.73 | 1.69 |
| IWP (g m$^{-2}$) | 9.34 | 11.2 | 0.05 | 0.08 | 0.02 | 0.07 | 0.08 | 0.09 |
| $r_e$ (μm) | 6.13 | 10.3 | -0.02 | -0.11 | 0.00 | -0.11 | -0.04 | -0.12 |



**Figures**

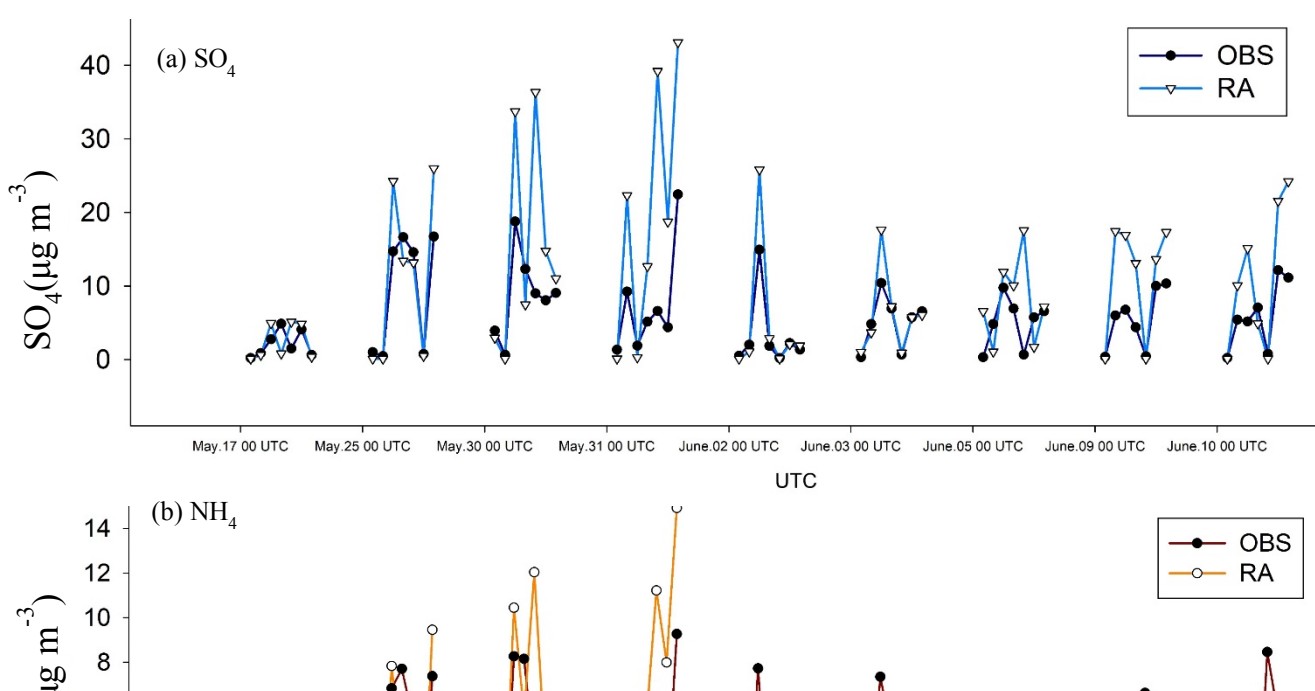

**Figure 1**. Time variation of the mass concentrations of (a) sulfate and (b) ammonium measured by the NASA DC-8 aircraft (OBS, black line) and simulated by AS (colored line)



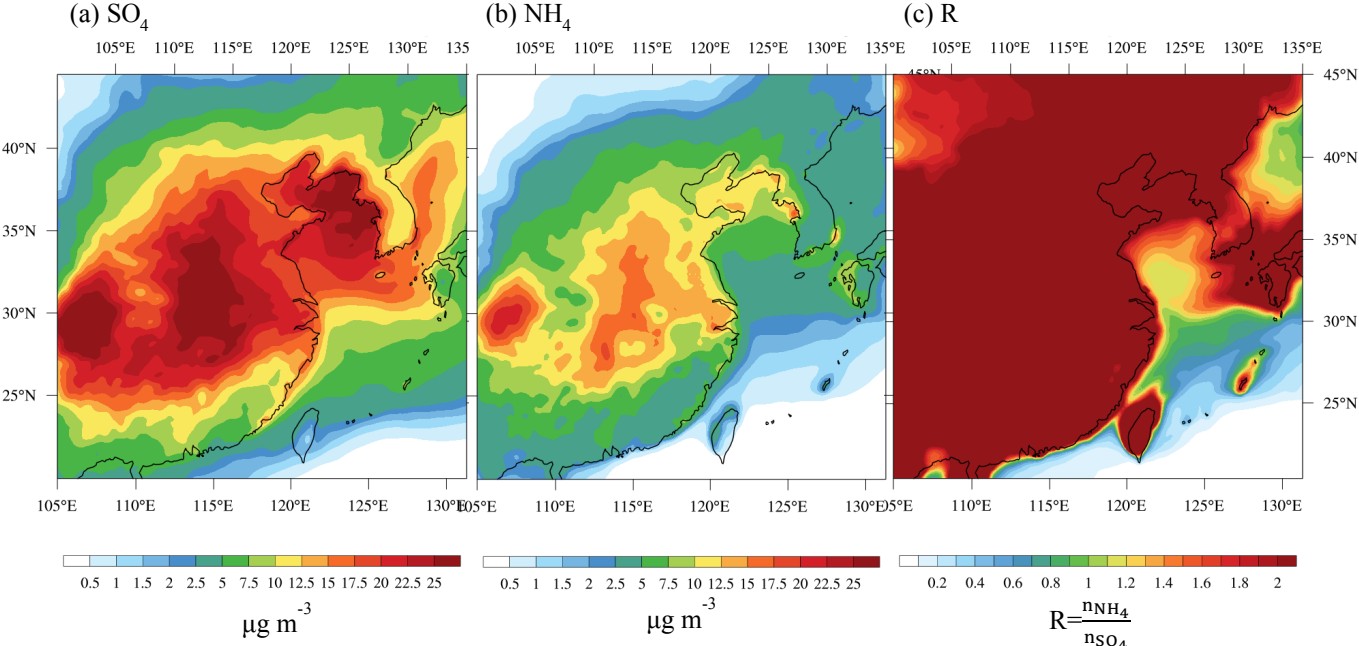

**Figure 2**. 30 days averaged (0000 UTC 16 May to 0000 UTC 15 June, 2016) spatial distribution of the mass concentrations of (a) sulfate and (b) ammonium, and (c) the molar ratio of ammonium to sulfate (R) at the surface, from AS.

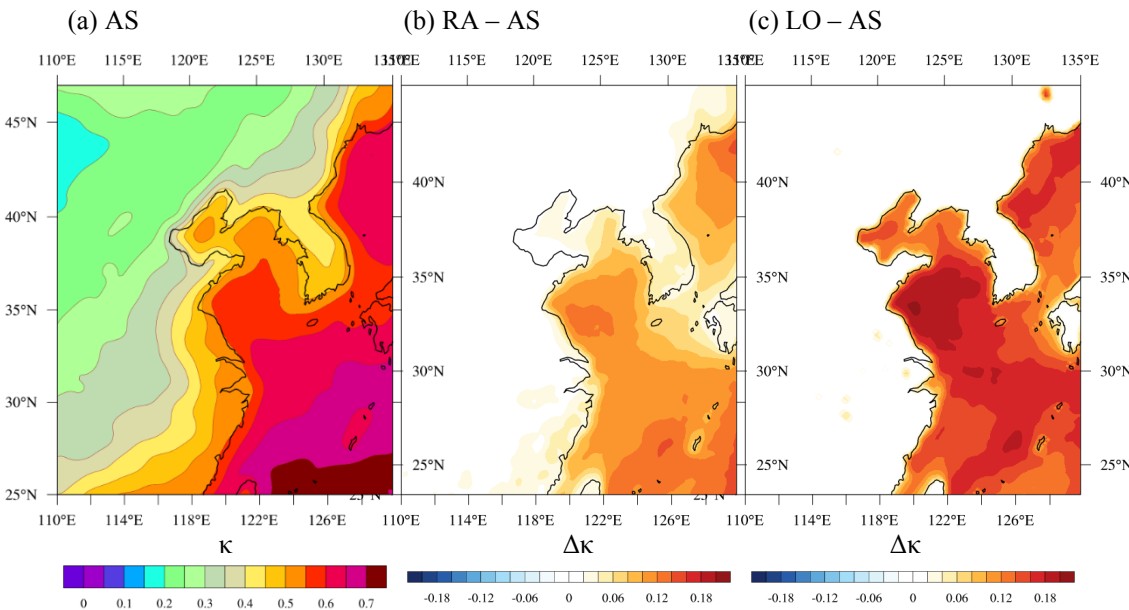

**Figure 3**. (a) Spatial distribution of the hygroscopicity parameter (κ) at the accumulation mode simulated by AS, the difference of κ (b) between RA and AS, and (c) between LO and AS, at the surface.

610





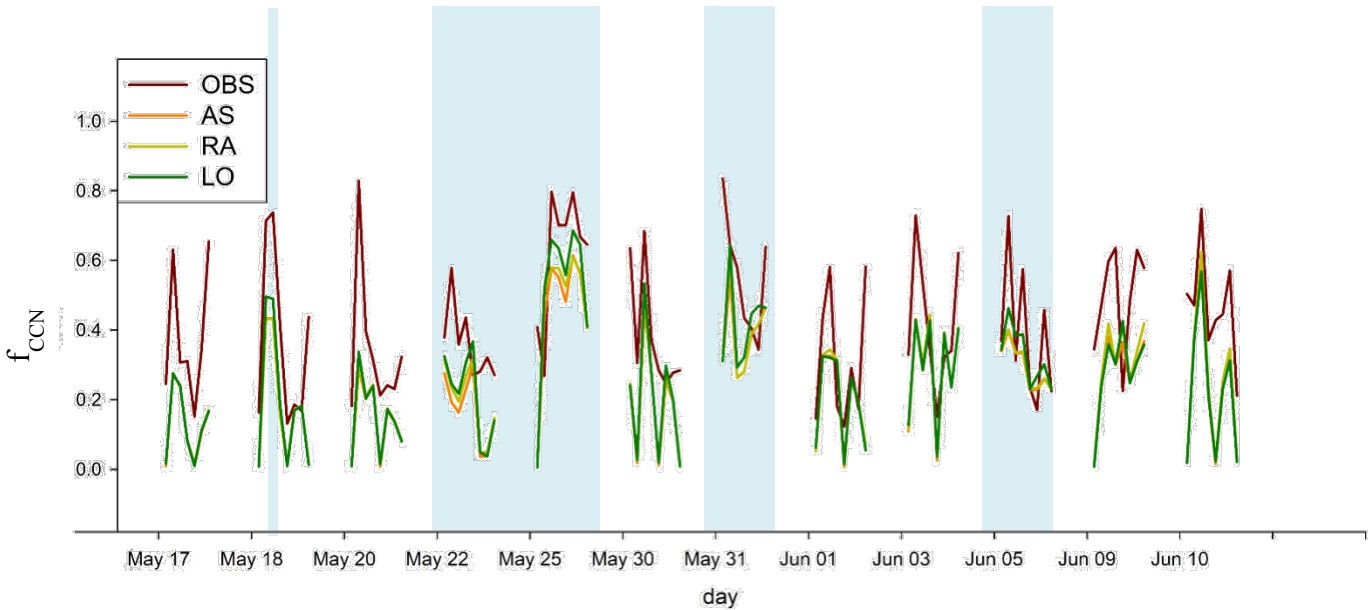

**Figure 4.** Time variation of the CCN activation fractions at 0.6% supersaturation ($f_{CCN}$) measured by the NASA DC-8 aircraft (OBS) and simulated by AS, RA, and LO. Blue shaded regions denote the time over the sea.





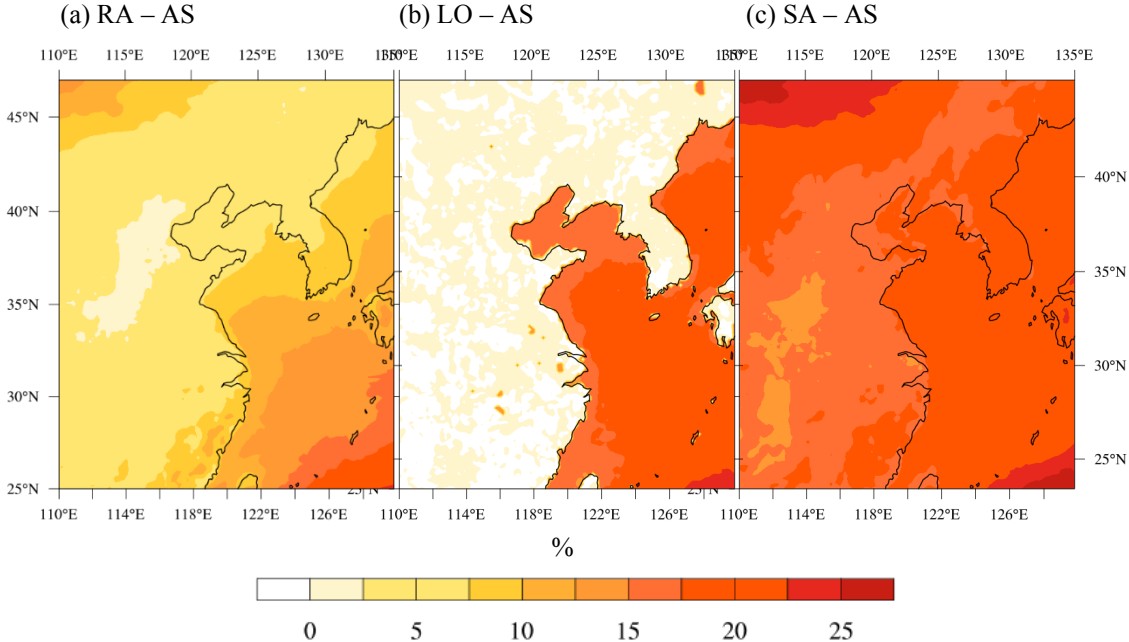

**Figure 5**. Percentage difference of column-integrated CCN activation fraction at 0.6% supersaturation (f$_{CCN}$) in AS and
615    sensitivity simulations, (a) RA – AS, (b) LO – AS, and (c) SA – AS.



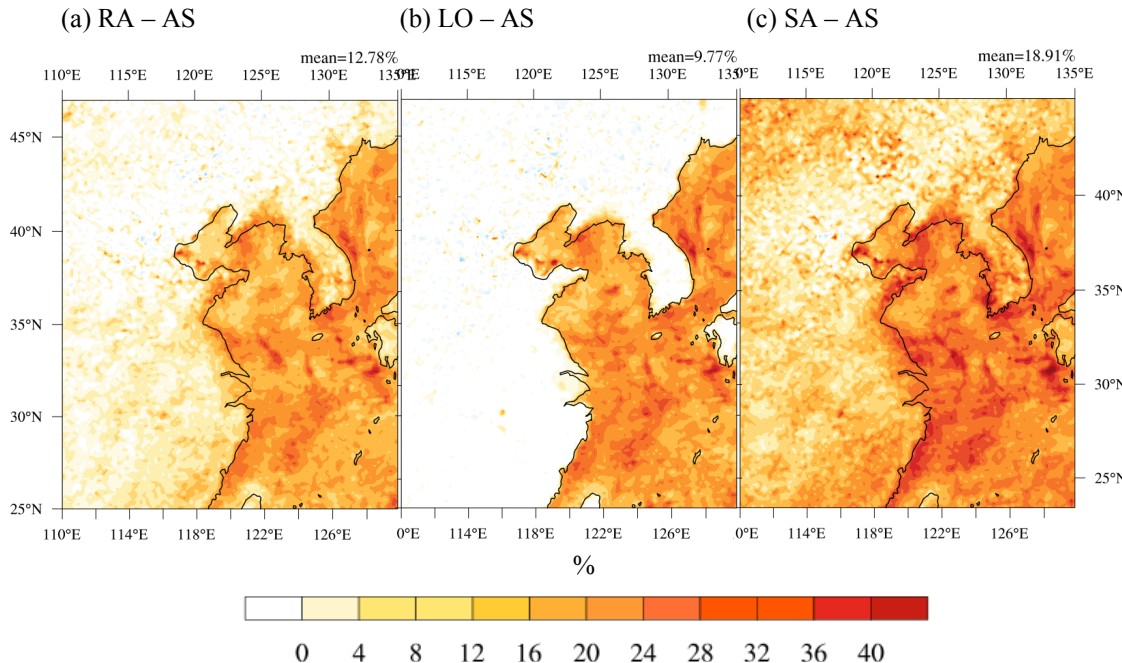

**Figure 6**. Same as Fig. 5 except for cloud droplet number concentration (CDNC).





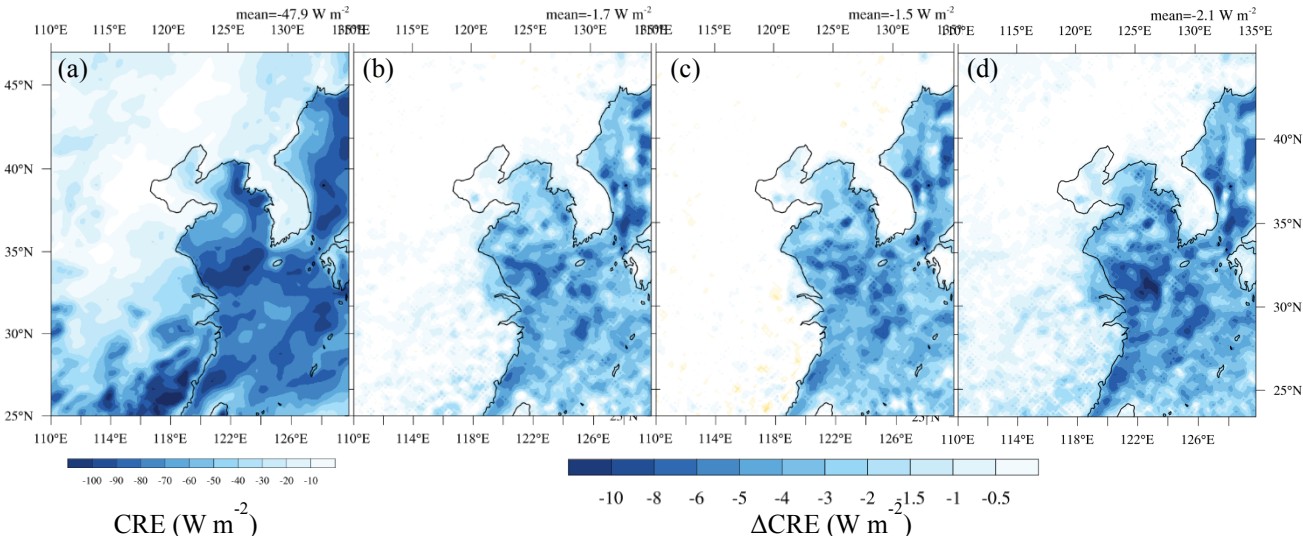

**Figure 7**. Simulated 30 days averaged (0000 UTC 16 May to 0000 UTC 15 June, 2016) cloud radiative effect (CRE) for (a)
620   AS and the differences (ΔCRE) between (b) RA and AS, (c) LO and AS, and (d) SA and AS.