# Peer review of "Optimization of Sulfate Aerosol Hygroscopicity Parameter in WRF-Chem"

_Geoscientific Model Development, 2020_

## Referee Comment (RC1) · Anonymous Referee #1 · 26 Aug 2020

The authors present model results of aerosol over East Asia using a new parameterization that accounts for varying sulfate particle species. The introduction provides a compelling argument as to why spatial and temporal variations in sulfate particle species should be accounted for in models. Furthermore, the analysis showing differing CCN activation fractions show the considerable contrast between the default and proposed methods. The manuscript is well structured, well written and the results are of great importance. I only have 1 question and a couple minor suggestions that I believe should be addressed before publication.

Section 2.4 - Why is KNH4HSO4 not also considered when R <2? Is there literature to support this exclusion?

Line 168 – I assume these are all boundary layer measurements. Please clarify in text.

[Figure]

Figure 1 – Consider adding shading to Figure 1 (similar to Figure 4) to identify measurements over land vs sea.

**[GMDD](https://doi.org/10.5194/gmd-2020-170)**

---

## Referee Comment (RC2) · Richard Moore (Referee) · 8 Sep 2020

Review of ``Optimization of Sulfate Aerosol Hygroscopicity Parameter in WRF-Chem version (3.8.1)'' by Kim et al.

The manuscript describes a new method for parameterizing the hygroscopicity of sulfate aerosols within WRF-Chem that accounts for whether the sulfate exists as neutralized ammonium sulfate or as sulfuric acid. A simple mixing rule is proposed based on the ratio of ammonium to sulfate ions, and multiple simulations are conducted assuming constant values for the hygroscopicity of each aerosol species. Model results are shown for southeast Asia and are compared to airborne observations from the KORUS-AQ field campaign. The model suggests that the sulfate aerosols over land are fully neutralized by ammonium ions, while the molar ratio of ammonium to sulfate is lower over coastal waters. This gives rise to significant changes in the hygroscopicity parameter over the water as well as notable changes in CCN, cloud droplet number, and cloud radiative effect over the entire domain. The comparisons to KORUS-AQ data show that the model captures the observed variability in aerosol mass; although, sulfate is significantly overpredicted during many time periods (Figure 1). The observed CCN-active aerosol fraction also tends to exceed model predictions, even when using the new parameterization (Figure 4). Overall, the manuscript is fairly well-written and the topic is interesting. A significant omission is not considering partial neutralization of the sulfate to form ammonium bisulfate. In addition, it is not well discussed how additional nitrate, sodium, or organic species from either continental or marine sources may impact the aerosol ion balance and acidity. I also note that the assumed hygroscopicity values do not appear consistent with what would be expected based on thermodynamic (Köhler) theory for CCN activation. Finally, some of the major conclusions about the "reliability" of the new parameterization need to be better supported by the results/discussion. The manuscript may be publishable, but only after the following comments are satisfactorily addressed.

**Specific comments:**

1. It seems appropriate to also consider ammonium bisulfate using simple mole balance mixing rules following Nenes et al. (1998) and Moore et al. (2011, 2012). Using the molar ratio of ammonium to sulfate ions, $R = n_{\mathrm{NH_4}}/n_{\mathrm{SO_4}}$,

    For $R \leq 1$,
    $$n_{\mathrm{(NH_4)_2SO_4}} = 0$$
    $$n_{\mathrm{NH_4HSO_4}} = n_{\mathrm{NH_4}}$$
    $$n_{\mathrm{H_2SO_4}} = n_{\mathrm{SO_4}} - n_{\mathrm{NH_4}}$$

    For $1 < R < 2$,
    $$n_{\mathrm{(NH_4)_2SO_4}} = n_{\mathrm{NH_4}} - n_{\mathrm{SO_4}}$$
    $$n_{\mathrm{NH_4HSO_4}} = 2n_{\mathrm{SO_4}} - n_{\mathrm{NH_4}}$$
    $$n_{\mathrm{H_2SO_4}} = 0$$

    For $R \geq 2$,
    $$n_{\mathrm{(NH_4)_2SO_4}} = n_{\mathrm{SO_4}}$$
    $$n_{\mathrm{NH_4HSO_4}} = 0$$
    $$n_{\mathrm{H_2SO_4}} = 0$$

where $n_{(NH_4)_2SO_4}$, $n_{NH_4HSO_4}$, and $n_{H_2SO_4}$ are the moles of ammonium sulfate, ammonium bisulfate, and sulfuric acid, respectively.

2. The chosen κ values of 0.53 and 1.19 for ammonium sulfate and sulfuric acid that are reported on Line 95 do not seem correct to me. From the Köhler Theory equations given by Seinfeld and Pandis (2016) and the approximation for κ given in Petters and Kreidenweis (2007), we know that

$$\ln^2 S_c = \frac{4A^3}{27\kappa D_d^3} = \frac{4A^3 \rho_w M_s}{27\nu \rho_s M_w D_d^3}$$

which yields

$$\kappa = \left(\frac{M_w}{\rho_w}\right)\left(\frac{\rho_s}{M_s}\right)\nu$$

where $M_w$ and $\rho_w$ are the molar mass and density of water, respectively; $M_s$ and $\rho_s$ are the molar mass and density of water of the dry solute, respectively; and $\nu$ is the van't Hoff factor that represents incomplete solute dissociation. Fortunately, these parameters are known for the sulfate-ammonium salts studied in this work, and the pure component van't Hoff factor can be calculated using a suitable thermodynamic model (e.g., the Pitzer model) to account for the supersaturation-dependent change in solute molality at the point of CCN activation.

[Figure]

Neglecting the supersaturation-dependence and taking an average of the calculated kappas over the 0.1-1% supersaturation range yields estimates for kappa of 0.60, 0.80, and 0.96 for ammonium sulfate, ammonium bisulfate, and sulfuric acid, respectively. These values are in reasonable agreement with the CCN-derived values reported by Petters and Kreidenweis (2007) of 0.61 and 0.90 for ammonium sulfate and sulfuric acid, respectively, and significantly different from the values reported in this manuscript.

3. On Line 90, it is stated that the upper bound of kappa is around 1.40, but this is not a theoretical limit. Rather, Petters and Kreidenweis (2007) note that this is the upper end of the range of typical hygroscopic species of atmospheric relevance. Please clarify or strike this sentence.

4. The first part of Equation 3 suggests that R could be negative, which is non-physical.

5. The discussion on Page 5 and the kappa values for land and sea assumed on Line 144 also seem unrealistic, especially for the coastal region studied here that is susceptible to significant transport of continental emissions. This line of reasoning seems to suggest that the source of aerosol SO4 is de-coupled from ammonia for the marine atmosphere and that there is a significant local marine source of SO4. What is the contribution of SO4 over the sea that is due to local sources versus transport?

6. How often is the ammonium data not available, and does this land-sea kappa parameterization differ notably from the R-based parameterization during periods where ammonium data are available?

7. On Line 193, it is stated that marine aerosols consist of mostly hydrophilic substances, which I'd suggest be revised to say "hygroscopic substances".

8. The citation for Park et al., 2020 that is referenced on Line 207 does not appear to be in the reference list.

9. For Figure 4 and related discussion on Lines 209-219, please also show the CCN and aerosol number concentration in $cm^{-3}$ in addition to the CCN-active fractions. How well do the AS, SA, RA, and LO simulations capture the observed variability in CCN and aerosol number concentrations measured during KORUS-AQ?

10. It's hard to see the differences in Figure 4 that accompany the discussion on Lines 220-227 that the new simulations outperform the AS simulation. The clearest discrepancies between the orange, yellow, and green curves appear during the May 22 – May 30 time period when the comparison is made over the ocean. Is this period driving the error metrics, and is the difference due to the model computed R values or because of the imposed land-sea kappas? Please add a table with the calculated error metrics that underpin these conclusions.

11. The statement on Lines 286-287 that the new parameterizations "could produce more reliable aerosol and CCN concentrations than the previous method" (and again on Line 300) is not currently well supported by the manuscript. First, the aerosol and CCN concentration simulation results need to be added to the manuscript (as requested in the comment above), and second, it's not clear that the results

are more or less reliable. The model results do seem to show significant differences across the simulations, but there needs to be more quantitative discussion about why one or the other simulation would be "more reliable".

12. What is the support for the statement on Lines 289-290 that the increased CDNCs "suppress local precipitation, prolong cloud lifetime, and consequently reflect more sunlight"? I agree that these processes may be plausible explanations for the simulated changes in CRE, but I don't think that the manuscript establishes a clear causal link to these processes.

13. What is the contribution of aerosol nitrate and sodium ions to the observations and simulations? How does the presence of these additional aerosol constituents impact the results shown here?

**References Cited:**

Nenes, A., Pandis, S. N., and Pilinis, C.: ISORROPIA: A new thermodynamic equilibrium model for multiphase multicomponent aerosols, Aquat. Geochem., 4, 123–152, doi:10.1023/A:1009604003981, 1998.

Moore, R. H., Bahreini, R., Brock, C. A., Froyd, K. D., Cozic, J., Holloway, J. S., Middlebrook, A. M., Murphy, D. M., and Nenes, A.: Hygroscopicity and composition of Alaskan Arctic CCN during April 2008, Atmos. Chem. Phys., 11, 11807–11825, https://doi.org/10.5194/acp-11-11807-2011, 2011.

Moore, R. H., Cerully, K., Bahreini, R., Brock, C. A., Middlebrook, A. M., and Nenes, A. (2012), Hygroscopicity and composition of California CCN during summer 2010, J. Geophys. Res., 117, D00V12, doi:10.1029/2011JD017352.

Petters, M. D. and Kreidenweis, S. M.: A single parameter representation of hygroscopic growth and cloud condensation nucleus activity, Atmos. Chem. Phys., 7, 1961–1971, https://doi.org/10.5194/acp-7-1961-2007, 2007.Pitzer, 1979

Seinfeld, J. H., & Pandis, S. N. (2016). Atmospheric chemistry and physics: From air pollution to climate change.

---

## Author Comment (AC1) · 7 Oct 2020

**Response to Referee 1**

Q1. Section 2.4 - Why is KNH4HSO4 not also considered when R<2? Is there literature to support this exclusion?

A1. As the referee mentioned, there are various chemical species in sulfate aerosols. Nevertheless, most state-of-the-art models consider sulfate aerosols as a single chemical species, either ammonium sulfate or sulfuric acid (see Section 2.3 in the original manuscript; i.e., the aerosol model in Unified Model (UM) considers that all sulfate aerosols are sulfuric acids (Mann et al., 2010), while ECHAM-HAM considers that all sulfate aerosols are ammonium sulfates (Zhang et al. 2012)). This is relevant to the fact that most atmospheric models treat chemical feedback in aerosol-cloud interactions with bulk physicochemical properties of aerosols. Even though some aerosol models are available to consider full chemistry feedbacks with clouds, they are not used in large scale atmospheric models to treat various sulfate aerosols because of the complexity of the chemical processes and high computational cost. For this reason, we propose simple parameters for representing two major sulfate aerosols that are considered in many atmospheric models.

Q2. Line 168 – I assume these are all boundary layer measurements. Please clarify in text.

A2. Yes, they are all boundary layer measurements. We add a line in the revised manuscript as below.

(Line 170) "The measurements are taken within the boundary layer."

Q3. Figure 1 – Consider adding shading to Figure 1 (similar to Figure 4) to identify measurements over land vs sea.

A3. Thanks. We add blue shadings for the measurements over the sea in Fig. 1 as is done in Fig. 4.

---

## Author Comment (AC2) · 7 Oct 2020

**Response to the Referee 2**

Q1. It seems appropriate to also consider ammonium bisulfate using simple mole balance mixing rules following Nenes et al. (1998) and Moore et al. (2011, 2012). Using the molar ratio of ammonium to sulfate ions, $R = n_{NH4} / n_{SO4}$ ,

For $R \leq 1$,

$n_{(NH4)2SO4} = 0$

$n_{NH4HSO4} = n_{NH4}$

$n_{H2SO4} = n_{SO4} - n_{NH4}$

For $1 < R < 2$,

$n_{(NH4)2SO4} = n_{NH4} - n_{SO4}$

$n_{NH4HSO4} = 2n_{SO4} - n_{NH4}$

$n_{H2SO4} = 0$

For $R \geq 2$,

$n_{(NH4)2SO4} = n_{SO4}$

$n_{NH4HSO4} = 0$

$n_{H2SO4} = 0$

where $n_{(NH4)2SO4}$, $n_{NH4HSO4}$, and $n_{H2SO4}$ are the moles of ammonium sulfate, ammonium bisulfate, and sulfuric acid, respectively.

A1. Thank you for your constructive comments. Your suggestion seems very legitimate. However, we think that to make additional model simulations incorporating your suggestion is too much to do at this point in time. Instead, we mention in the revised manuscript that to be more realistic, ammonium bisulfate may also need to be considered in such a way that you described in your comment.

(Line 126–130) "*To be more realistic, ammonium bisulfate may also need to be considered: when the number of $SO_4^{2-}$ is smaller than $NH_4^+$, the sulfates appear as a mixture of ammonium bisulfates and sulfuric acids, and when the number of $SO_4^{2-}$ is greater than $NH_4^+$ but not twice as large as $NH_4^+$, the sulfates appear as a mixture of ammonium bisulfates and ammonium sulfates (Nenes et al., 1998; Moore et al., 2011, 2012). For simplicity, however, such partitioning is not considered in this study.*"

Q2. The chosen κ values of 0.53 and 1.19 for ammonium sulfate and sulfuric acid that are reported on Line 95 do not seem correct to me. From the Köhler Theory equations given by

Seinfeld and Pandis(2016) and the approximation for $\kappa$ given in Petters and Kreidenweis (2007), we know that

$$\ln^2 S_c = \frac{4A^3}{27\kappa D_d^3} = \frac{4A^3 \rho_w M_s}{27\nu\rho_s W_w D_d^3}$$

which yields

$$\kappa = \left(\frac{M_w}{\rho_w}\right)\left(\frac{\rho}{M_s}\right)\nu$$

where $M_w$ and $\rho_w$ are the molar mass and density of water, respectively; $M_s$ and $\rho_s$ are the molar mass and density of water of the dry solute, respectively; and $\nu$ is the van't Hoff factor that represents incomplete solute dissociation. Fortunately, these parameters are known for the sulfate-ammonium salts studied in this work, and the pure component van't Hoff factor can be calculated using a suitable thermodynamic model (e.g., the Pitzer model) to account for the supersaturation-dependent change in solute molality at the point of CCN activation. Neglecting the supersaturation-dependence and taking an average of the calculated kappas over the 0.1- 1% supersaturation range yields estimates for kappa of 0.60, 0.80, and 0.96 for ammonium sulfate, ammonium bisulfate, and sulfuric acid, respectively. These values are in reasonable agreement with the CCN-derived values reported by Petters and Kreidenweis (2007) of 0.61 and 0.90 for ammonium sulfate and sulfuric acid, respectively, and significantly different from the values reported in this manuscript.

A2. Thank you for your legitimate comment. The chosen $\kappa$ values of 0.53 and 1.19 for ammonium sulfate and sulfuric acid are the HTDMA-derived values from Petters and Kreidenweis (2007). In the context of cloud droplet activation, CCN-derived $\kappa$ values might be more appropriate to use as you suggested because they were measured under cloudy (i.e, supersaturated) condition. In this study, we try to manifest the effect of the different $\kappa$ values of the two major sulfate species and that is the main reason for choosing HTDMA-derived $\kappa$ values that show greater difference between ammonium sulfate and sulfuric acid, instead of CCN-derived ones that show smaller difference. We add statements which explain that the CCN-derived $\kappa$ values can also be used and if so, what could be the expected results.

(Line 302–306) "*To note is that the $\kappa$ values of 0.53 and 1.19 for $(NH_4)_2SO_4$ and $H_2SO_4$ that we used in this study were HTDMA-derived, instead of CCN-derived, which were 0.61 and 0.90, respectively (Petters and Kreidenweis, 2007). If CCN-derived $\kappa$ values were used, CDNC would generally have decreased because $\kappa$ became lower and the contrast between $(NH_4)_2SO_4$ and $H_2SO_4$ would have been decreased to a certain degree.*"

Q3. On Line 90, it is stated that the upper bound of kappa is around 1.40, but this is not a theoretical limit. Rather, Petters and Kreidenweis (2007) note that this is the upper end of the range of typical hygroscopic species of atmospheric relevance. Please clarify or strike this sentence.

A3. Thank you for your correction. We modified the sentence in the revised manuscript.

(Line 91) "*The upper end of the κ value for hygroscopic species of atmospheric relevance is around 1.40 (Petter and Kreidenweis, 2007).*"

Q4. The first part of Equation 3 suggests that R could be negative, which is non-physical.

A4. Thanks. We modified the first part of Equation 3 to accommodate your comment.

Q5. The discussion on Page 5 and the kappa values for land and sea assumed on Line 144 also seem unrealistic, especially for the coastal region studied here that is susceptible to significant transport of continental emissions. This line of reasoning seems to suggest that the source of aerosol SO4 is decoupled from ammonia for the marine atmosphere and that there is a significant local marine source of SO4. What is the contribution of SO4 over the sea that is due to local sources versus transport?

A5. What is discussed in the later part of Page 5 is basically for LO parameterization, which can be used when ammonium information is not available. Simply, κ for land grid points is assumed to be 0.53 and that for ocean grid points to be 1.19. For the coastal grid points, we use Eq. (4). Indeed, because of the intrusion of maritime air mass over the coastal land and transportation of continental pollutants over the coastal sea, κ values of RA and LO showed differences in the coastal regions as described in 4.2 because RA did consider the effect of air mass exchanges but LO did not. So for LO simulation, it can be said that the source of aerosol $SO_4$ is decoupled from ammonia for the marine atmosphere. Note that the κ distribution shown in Fig. 3 are for all aerosols that of course include sulfate species. It is estimated that the dimethyl-sulfide (DMS)—chemicals produced by phytoplankton—take approximately 45% and 18% of the total sulfate column burden in the Southern and Northern Hemisphere, respectively (Gondwe et al., 2003; Kloster et al., 2006). Therefore, DMS is the main source of sulfate aerosols over the sea. As you mentioned, $SO_4$ may also be transported from land. However, it is difficult to estimate the contribution of $SO_4$ over the sea that is due to local sources and transport in the model we used in this study.

Q6. How often is the ammonium data not available, and does this land-sea kappa parameterization differ notably from the R-based parameterization during periods where ammonium data are available?

A6. The sentence "the ammonium information is not available" (Line 147 in the original manuscript) indicates the situation when the atmospheric models do not have ammonium ancillary files or do not calculate ammonium chemical processes. In fact, most state-of-the-art atmospheric models consider only few representative chemical species of aerosols such as sulfate, black carbon, organic carbon, sea salt, and dust but no ammonium (e.g., Mann et al., 2010, Zhang et al. 2012). This is the reason why we alternatively suggest using LO parameterization for the models that do not provide ammonium data. Yes, RA and LO simulations do show clear differences as described in 4.2.

Q7. On Line 193, it is stated that marine aerosols consist of mostly hydrophilic substances, which I'd suggest be revised to say "hygroscopic substances".

A7. The word 'hydrophilic' is replaced by 'hygroscopic' as suggested. (Line 197).

Q8. The citation for Park et al., 2020 that is referenced on Line 207 does not appear to be in the reference list.

A8. The citation for Park et al. (2020) appeared in the original manuscript but you somehow missed it. Since the full citation information is now available for this reference, the citation is slightly modified (Line 539).

Q9. For Figure 4 and related discussion on Lines 209-219, please also show the CCN and aerosol number concentration in cm-3 in addition to the CCN-active fractions. How well do the AS, SA, RA, and LO simulations capture the observed variability in CCN and aerosol number concentrations measured during KORUS-AQ?

A9. Aerosol number concentration distribution in AS, SA, RA, and LO are almost the same because we did not alter any of the chemical processes except for sulfate hygroscopicity. CCN distribution in Fig. 4 format is shown in Fig. A1 below. In the model, CCN concentration is calculated from $f_{CCN}$ and aerosol concentration by multiplying the two. Moreover, since the aerosol number concentration in all these simulations are similar, $f_{CCN}$ distribution can be considered almost as a proxy for CCN number concentration distribution. Therefore it seems not really necessary to show CCN distribution, too, in the manuscript. So instead of showing CCN number concentration distributions, we added discussion on the averages of aerosol and CCN number concentrations and how well the model simulations captured the observed variability of CCN concentration measured during KORUS-AQ in the revised manuscript as below.

(Line 214–222) *"However, $f_{CCN}$ are underestimated mainly due to the underestimation of CCN concentrations. The average aerosol (CN) number concentrations for the flight track in all simulations (AS, SA, RA, and LO) and the actual observed values during the flight are 5934 $cm^{-3}$ and 5794 $cm^{-3}$, respectively. So unlike Georgiou et al.(2018) that showed that the WRF-Chem coupled with MADE/SORGAM tended to overestimate aerosol number concentrations, our simulations only slightly overestimated aerosol number concentrations. The average CCN number concentration at 0.6% supersaturation for AS, RA and LO simulations are 982 $cm^{-3}$, 1027 $cm^{-3}$, 1057 $cm^{-3}$, respectively, but the observation was 2154 $cm^{-3}$. Such underestimated CCN concentrations seem to be due to the systematic error in WRF-Chem. As discussed in Tuccella et al. (2015), the uncertainty of updraft velocity parameterization and bulk hygroscopicity of aerosols lead to underestimation of CCN concentration and CCN efficiency (CCN/CN) by a factor of 1.5 and 3.8, respectively. Nevertheless, over land, AS, ..."*

(Line 229–232) *"Simulated $f_{CCN}$ in RA has a high spatiotemporal correlation with the observation over the Yellow Sea (i.e., 0.83), while AS shows a rather lower correlation (i.e., 0.65). Such difference stems from the fact that the R values (molar ratio of ammonium to sulfate) vary significantly over the Yellow sea due to the transportation of anthropogenic chemicals by*

*westerlies and such variability is taken into account in RA. This improvement highlights the importance of appropriate chemical representation in atmospheric models."*

[Figure]

Figure A1. Time variation of CCN concentration at 0.6% supersaturation measured by aircraft (OBS) and simulated in AS, RA, and LO.

Q10. It's hard to see the differences in Figure 4 that accompany the discussion on Lines 220-227 that the new simulations outperform the AS simulation. The clearest discrepancies between the orange, yellow, and green curves appear during the May 22 – May 30 time period when the comparison is made over the ocean. Is this period driving the error metrics, and is the difference due to the model computed R values or because of the imposed land-sea kappas? Please add a table with the calculated error metrics that underpin these conclusions.

A10. Yes, the oceanic period mainly drove the error metrics and the difference is due to model computed/imposed kappa values for sulfates. In the revised manuscript, Table 3 shows the model performance statistics. The paragraph that explains Table 3 is also rewritten in the revised manuscript (Line 235-241). Further support for these conclusions can also be found in A9.

Table 3. Values of the three criteria suggested in Kumar et al. (1993).

| | AS | RA | LO |
|---|---|---|---|
| NMSE < 0.5 | 0.53 | 0.48 | 0.43 |
| -0.5 < Fractional Bias $(= 2 \times \frac{\overline{obs} - \overline{sim}}{\overline{obs} + \overline{sim}})$ <0.5 | 0.54 | 0.5 | 0.46 |
| 0.5 < Ratio $(= sim/obs)$ < 2 | 0.59 | 0.65 | 0.65 |

Q11. The statement on Lines 286-287 that the new parameterizations "could produce more reliable aerosol and CCN concentrations than the previous method" (and again on Line 300) is not currently well supported by the manuscript. First, the aerosol and CCN concentration simulation results need to be added to the manuscript (as requested in the comment above), and second, it's not clear that the results are more or less reliable. The model results do seem to

show significant differences across the simulations, but there needs to be more quantitative discussion about why one or the other simulation would be "more reliable".

A11. We think that what we discussed in A9 can also be the response to this comment.

Q12. What is the support for the statement on Lines 289-290 that the increased CDNCs "suppress local precipitation, prolong cloud lifetime, and consequently reflect more sunlight"? I agree that these processes may be plausible explanations for the simulated changes in CRE, but I don't think that the manuscript establishes a clear causal link to these processes.

A12. The supporting data for this statement are from Tables 1 and 2 and sub-sections 4.4 and 4.5. Table 1 shows that CDNC is lower in AS than in all other simulations especially over the ocean. Meanwhile, compared to AS, RA and LO simulate less rainwater, more LWP, and smaller effective radii, especially over the ocean, as shown in Table 2. Such contrasts imply suppression of precipitation for increased CDNC. Cloud lifetime effect is difficult to quantify, but increased LWP and reduced effective radii in RA and LO, compared to AS, may implicitly suggest cloud life time effect for increased CDNC. These are discussed in sub-sections 4.4 and 4.5.

Q13. What is the contribution of aerosol nitrate and sodium ions to the observations and simulations? How does the presence of these additional aerosol constituents impact the results shown here?

A13. In this study, we focused on the effect of sulfate aerosols because they are known to explain a majority (64%) of the radiative forcing from aerosol-cloud interaction (Zelinka et al., 2014). We only made changes in the representation of sulfate aerosol species and did not alter any other chemical processes, and we found that the amount of nitrate and sea salt aerosols in AS, RA, and LO simulations were similar. Perhaps this implies that different treatment of sulfate aerosols did not significantly affect nitrate and sea salt aerosols. However, it is difficult to estimate how the presence of nitrate and sea salt aerosols impacted the results in our simulations. So in the revised manuscript, we only add discussion on the importance of nitrate and sea salt aerosols as below.

(Line 331–338) *"In this study we did not discuss other important aerosol species. For instance, the proportion of mass concentrations of nitrate ions are almost as large as sulfate ions (Zhang et al., 2007; Moore et al., 2012), and nitrate also have spatio-temporally varying hygroscopicity due to the complex chemical reactions with other chemicals, i.e., ammonium, sodium, and calcium. In this study, we only made changes in the representation of sulfate aerosol species and did not alter any other chemical processes, and we find that the amount of nitrate and sea salt aerosols in AS, RA, and LO simulations were similar. Perhaps this implies that different treatment of sulfate aerosols did not significantly affect nitrate and sea salt aerosols. However, it is difficult to estimate how the presence of nitrate and sea salt aerosols impacted the results in our simulations. Future study may need to address such important issue in more detail."*

**Reference**

Gondwe, M., Krol, M., Gieskes, W., Klaassen, W., de Baar, H.: The contribution of ocean-leaving DMS to the global atmospheric burdens of DMS, MSA, $SO_2$, and NSS $SO_4$. *Global Biogeochemical Cycles*, **17**; DOI:10.1029/2002GB001937, 2003.

Kloster, S., Feichter, J., Maier-Reimer, E., Six, K.D., Stier, P., Wetzel, P.: DMS cycle in the marine ocean atmosphere system–a global model study. *Biogeosciences*, **3**, 29–51; DOI:10.5194/bg-3-29-2006, 2006.

Mann, G. W., Carslaw, K. S., Spracklen, D. V., Ridley, D. A., Manktelow, P. T., Chipperfield, M. P., Pickering, S. J., and Johnson, C. E.: Description and evaluation of GLOMAP-mode: a modal global aerosol microphysics model for the UKCA composition-climate model, *Geosci. Model Dev.,* **3**, 519–551, https://doi.org/10.5194/gmd-3-519-2010, 2010.

Moore, R. H., Bahreini, R., Brock, C. A., Froyd, K. D., Cozic, J., Holloway, J. S., Middlebrook, A. M., Murphy, D. M., and Nenes, A.: Hygroscopicity and composition of Alaskan Arctic CCN during April 2008, Atmos. Chem. Phys., 11, 11807–11825, https://doi.org/10.5194/acp-11-11807-2011, 2011.

Moore, R. H., Cerully, K., Bahreini, R., Brock, C. A., Middlebrook, A. M., and Nenes, A.: Hygroscopicity and composition of California CCN during summer 2010, J. Geophys. Res., 117, D00V12, doi:10.1029/2011JD017352, 2012.

Nenes, A., Pandis, S. N., and Pilinis, C.: ISORROPIA: A new thermodynamic equilibrium model for multiphase multicomponent aerosols, Aquat. Geochem., 4, 123–152, doi:10.1023/A:1009604003981, 1998.

Tuccella, P., Curci, G., Grell, G. A., Visconti, G., Crumeyrolle, S., Schwarzenboeck, A., and Mensah, A. A.: A new chemistry option in WRF-Chem v. 3.4 for the simulation of direct and indirect aerosol effects using VBS: evaluation against IMPACT-EUCAARI data, *Geosci. Model Dev.*, **8**, 2749–2776, https://doi.org/10.5194/gmd-8-2749-2015, 2015.

Zhang, K., O'Donnell, D., Kazil, J., Stier, P., Kinne, S., Lohmann, U., Ferrachat, S., Croft, B., Quaas, J., Wan, H., Rast, S., and Feichter, J.: The global aerosol-climate model ECHAM-HAM, version 2: sensitivity to improvements in process representations, *Atmos. Chem. Phys.*, **12**, 8911–8949, https://doi.org/10.5194/acp-12-8911-2012, 2012.